# Accurate and Low-Power Ultrasound–Radiofrequency (RF) Indoor Ranging Using MEMS Loudspeaker Arrays

**DOI:** 10.3390/s23187997

**Published:** 2023-09-20

**Authors:** Chesney Buyle, Lieven De Strycker, Liesbet Van der Perre

**Affiliations:** KU Leuven, WaveCore, Department of Electrical Engineering (ESAT), Ghent Technology Campus, 9000 Ghent, Belgium; lieven.destrycker@kuleuven.be (L.D.S.); liesbet.vanderperre@kuleuven.be (L.V.d.P.)

**Keywords:** indoor positioning, ultrasound, phased loudspeaker array, MEMS, low power

## Abstract

Accurately positioning energy-constrained devices in indoor environments is of great interest to many professional, care, and personal applications. Hybrid RF–acoustic ranging systems have shown to be a viable technology in this regard, enabling accurate distance measurements at ultra-low energy costs. However, they often suffer from self-interference due to multipaths in indoor environments. We replace the typical single loudspeaker beacons used in these systems with a phased loudspeaker array to promote the signal-to-interference-plus-noise ratio towards the tracked device. Specifically, we optimize the design of a low-cost uniform planar array (UPA) through simulation to achieve the best ranging performance using ultrasonic chirps. Furthermore, we compare the ranging performance of this optimized UPA configuration to a traditional, single-loudspeaker system. Simulations show that vertical phased-array configurations guarantee the lowest ranging errors in typical shoe-box environments, having a limited height with respect to their length and width. In these cases, a P50 ranging error of around 3 cm and P95 ranging error below 30 cm were achieved. Compared to a single-speaker system, a 10 × 2 vertical phased array was able to lower the P80 and P95 up to an order of magnitude.

## 1. Introduction

Accurate, low-power asset tracking plays an important role in the digitization of professional care and personal environments. Time-based acoustic positioning systems are interesting candidates in this regard as sound waves inherently exhibit a low propagation speed (∼343 m/s). Consequently, devices do not have to operate at high processing frequencies to establish accurate timing and thus distance measurements. This also benefits the energy consumption and ultimately the cost of the typically battery-powered tracked device. In this work, we focus on a system that adds a radio frequency (RF) channel next to the acoustic channel, forming a hybrid system. The RF signal serves as a time reference during the acoustic time-of-flight (TOF) measurements. Examples of such hybrid RF–acoustic positioning systems are the Active Bat Local Positioning System [1] and the Cricket System [2]. While the additional RF channel increases the hardware complexity of the tracked device, it can also provide a means for high-speed data transfer if needed. In this context, RF backscatter approaches have been proposed to further enhance the nodes’ autonomy [3,4].

On the other hand, time-based acoustic ranging measurements are susceptible to changes in ambient conditions and reflections induced by the environment. Parameters such as temperature and humidity can significantly alter the propagation speed of acoustic waves, affecting the TOF measurements [5,6]. Nevertheless, these fluctuations can be rather easily accounted for through additional low-cost sensors monitoring of the environmental characteristics. Multipath and scattering are the main sources of interference in typical room environments. Both signaling and physical approaches have been proposed to counter these effects. At the signaling level, narrowband pulses were traded for broadband signals, and pulse compression techniques have been adopted to increase the effective signal bandwidth [7,8]. Physical measures include the introduction of transducer arrays and beamforming techniques to promote the signal-to-noise ratio (SNR) of signals. While this latter approach has not been applied in time-based ranging systems, it is very common in, for example, sound source localization [9] or speech-enhancement applications [10], typically in the form of microphone arrays.

### 1.1. Related Work

Since the development of the first ultrasound positioning systems at the end of the past century, such as the Active Bat Local Positioning System [1] and the Cricket System [2], many works have focused on signal design and signal processing to further progress their performance. As these early systems relied on short, single-frequency pulses and basic energy thresholding to detect their arrival at the receiver, they showed low immunity to in-band noise and limited positioning accuracy on the order of decimeters. Moreover, these systems were not optimized to work in a multi-user environment, and their update rates were rather low [11]. A solution to these drawbacks was found in radar systems by means of spread spectrum techniques. These not only allowed the effective transmitted power to be increased for better SNR but also enabled high compression in the time domain via cross-correlation. This resulted in both a good target detection and ranging resolution. The works of [12,13] were among the first to introduce broadband signals in ultrasound positioning systems. The authors modulated 511-bit Gold codes on a 50 kHz carrier through binary phase-shift keying at a bit rate of 20 kHz. In [12], they established a test setup in a small office environment, where eight receivers were mounted on the ceiling. A set of synchronized transmitters emitted the Gold code-spreaded signals, and 3D position estimations were calculated based on TOF measurements. Even in a noisy environment, they reported 3D positioning accuracies slightly exceeding 2 cm. In [13], the same authors proposed a privacy-oriented broadband positioning system, with errors below 5 cm. Similar system designs were presented shortly after, which, for example, used smaller Gold codes at higher bit rates to obtain shorter transmissions [14]. Other works used different unitary spreading pseudorandom codes such as m-sequences [15] or Kasami codes [16]. Since the development of these broadband ultrasound positioning systems in the early 2000s, many efforts have been made in designing spreading sequences that are able to deal with strong multipath, the near-far effect, Doppler shifts and multiple access, among others. Pseudorandom sequences like Loosely Synchronous Sequences [17] and Complementary Sets of Sequences [18] were adopted in this context. More recent works have shown interest in polyphase sequences, such as Zadoff-Chu Codes, as they offer flexibility in length, show a good performance in periodic systems, and a good tolerance to Doppler shifts [19,20].

While in hybrid RF–acoustic positioning systems, the RF link is often used to provide a timing reference during TOF, it can also be used to support acoustic positioning in challenging scenarios. More specifically, acoustic ranging often suffers in places where no line of sight is available or strong multipath is present, for example, in densely obstructed environments. Consequently, researchers have proposed systems that use the radio link to a larger extent. In [21], the authors combined acoustic positioning with Wi-Fi fingerprinting and MEMS sensors, which performs better in the aforementioned circumstances. Experimental investigations showed that meter-level accuracies could be achieved in non-line-of-sight scenes. In [22], Bluetooth Low Energy (BLE) received signal strength measurements were combined with a time difference of arrival (TDoA) ultrasonic localization system to improve accuracy and robustness in fringe parts of the localization area.

Next to signal design, signal processing, and the aforementioned hybrid systems, physical measures like transducer arrays have been introduced to further progress acoustic localization. Sound source localization is a very common application in this regard, which involves determining the spatial direction or location of sources in the environment [9]. Moreover, it is the starting point of other notable applications such as sonar, speed enhancement, simultaneous localization and mapping (SLAM), and robot navigation [10,23]. Based on the field of sound source localization, transducer arrays typically appear in the form of microphone arrays. MUSIC, ESPRIT, GCC-PHAT, and beamforming are widely used techniques to determine the direction of arrival (DoA) of received signals [24]. Through beamforming, for example, sonar is able to construct one or more directional beams simultaneously to perform fine-grained target detection [25]. While loudspeaker arrays have been applied in a variety of fields, such as haptic control, object levitation, medical examination, and non-destructing testing, their application in acoustic localization has been very limited so far. Sonar systems have also been demonstrated using loudspeaker arrays for environmental mapping [26,27]. However, these systems do not focus on the localization of specific trackers in space. In sound zoning and binaural audio reproduction applications [28,29], loudspeakers arrays have been used to provide individual audio to spatially confined regions in the same room, but this strategy has not been applied for localization purposes.

### 1.2. Outlining Previous Work and the Research Gap

In recent work, our research team has proposed a chirp-based ultrasound ranging system tailored towards very low-power mobile devices [30]. After all, a very low energy consumption at the tracked device limits the battery cost and/or prolongs the device’s autonomy. We performed experimental measurements in a reverberant room using the aforementioned ultra-low-power chirp-based ranging technique and made three main observations [31]: (I) the error increased with distance, which could be attributed to signal attenuation and lower signal-to-noise ratios; (II) larger errors could be found closer to the edges of the room, which was potentially attributed to the high reverberation time of the room; and (III) low ranging errors could be observed within the area covered via the directional speaker pattern. These observations suggest that transducer arrays and beamforming techniques can be employed to further improve the ranging accuracy, as will be shown in this work. Similar to the concept of sonar, directional beams can be directed towards the tracked device. This not only increases the signal power at the tracked device but also limits multipath interference by avoiding transmissions in unwanted directions, thus increasing the signal-to-interference-plus-noise ratio (SINR). Since the energy and budget of a tracked device are often limited, it is practically more economical to keep the power-intensive acoustic transmissions on the infrastructure side. Consequently, this work considers phased loudspeaker arrays as opposed to the many microphone array designs used in acoustic localization.

### 1.3. Contributions

In this work, we replace the single-speaker beacon used in conventional time-based RF–acoustic ranging systems with a phased loudspeaker array. Specifically, we optimize the design of a low-cost uniform planar array (UPA) through simulations in order to achieve the best ranging performance in medium-sized rooms. The design is fitted to the state-of-the-art ranging scheme from [30] that enables accurate and energy-efficient TOF measurements through the pulse compression of ultrasonic chirps. In summary, the following two main research questions are investigated:Which UPA design leads to the best ranging accuracy over an entire room? Since a broadband chirp signal is emitted for TOF ranging, we first investigate at which operating frequency the loudspeaker array should be designed. In this regard, the beam patterns of multiple loudspeaker arrays with different interelement spacing are investigated. Hereafter, multiple array designs that are different in number of speakers and configuration are evaluated in terms of ranging accuracy over several shoe-box rooms. These medium-sized rooms differ in terms of dimensions to avoid optimization toward one specific room.How does the best-performing loudspeaker UPA, following the results of research question 1, perform with respect to a conventional single-speaker system? In this case, the cumulative distribution functions of the ranging errors of both systems are compared.

Extensive simulations show that vertical phased array configurations, designed at the center frequency of the emitted chirp, lead to the lowest ranging errors in typical shoe-box environments, where the height of the room is smaller than its length and width. In these cases, a P50 ranging error of around 3 cm and P95 ranging error below 30 cm are achieved. Compared to a single-speaker system, a 10 × 2 vertical phased array is able to lower the P80 and P95 up to an order of magnitude.

### 1.4. Assumptions

While a phased array improves the SNR of an acoustic signal in the main lobe direction, there is, however, one hurdle to overcome. During initial access, the infrastructure is unaware of the location of the tracked device in the environment. In other words, the phased array beacon must first determine its angle relative to this device. An RF direction-of-arrival system [32] could, for example, be employed to determine this information via the RF signals received from the tracked device. In this work, we further abstract how this angle information is obtained. Once azimuth and elevation angles are known, the directional acoustic ranging measurements can be carried out. The obtained position can then be used to further converge toward the real position. Furthermore, we show the performance of the proposed concept through static devices only. In future work, one can build upon this framework to further investigate mobile devices. They bring another level of complexity to the positioning system as a continuous redirection of the main lobe is required.

## 2. Ultra-Low-Power Acoustic–RF Ranging

The acoustic–RF ranging scheme built upon in this paper is based on the work of [30] and is summarized in Figure 1. Consider a beacon *B* equipped with a single loudspeaker. The beacon is controlled on the infrastructure side and has no firm restrictions on power and energy. Its target is to measure the distance Δx to a (mobile) energy-constrained device *T* positioned at an unknown location. When no ranging measurement is carried out, *T* remains in a low-power sleep mode. Beacon *B* initiates an acoustic ranging measurement by transmitting an ultrasonic chirp with length $\Delta t_{\mathrm{chirp}}$ and bandwidth Δf. Once the entire chirp is transmitted, *B* broadcasts an RF signal to wake up the tracked device *T*. Upon reception of the RF signal, *T* wakes up immediately and starts sampling the still-propagating ($v_{\mathrm{RF}}\gg v_{\mathrm{sound}}$) chirp signal for a time window $\Delta t_{RX}$, a fraction of $\Delta t_{\mathrm{chirp}}$. Once the sampling process is complete, it sends the captured part of the ultrasonic chirp back to the beacon over RF and goes back to sleep. Depending on the distance Δx between both entities, *T* will have received a specific fragment of the ultrasonic chirp. In other words, the frequency information captured in the acoustic fragment at *T* reveals its distance to the beacon. By means of the cross-correlation (CC) between the captured chirp fragment and the fully transmitted ultrasonic chirp signal, beacon *B* is able to calculate the time Δt it took for this specific part of the chirp to propagate to *T*.

In Figure 1, beacon *B* transmits the ultrasonic chirp starting with the lower frequencies and ending with the higher frequencies. If the device *T* had been positioned right in front of the beacon, the captured fragment by *T* would contain the highest chirp frequencies. The larger the distance Δx, the lower the frequencies that will be received. Consequently, in order for *B* to calculate the time delay Δt, it shifts the captured fragment during CC starting at the higher frequencies of the full chirp signal to the lower frequencies. The time shift corresponding to the maximum of the CC signal ideally gives Δt. Assuming that the speed of sound $v_{\mathrm{sound}}$ is frequency-independent, the distance $\Delta x=v_{\mathrm{sound}}\Delta t$ can be calculated.

The acoustic–RF ranging scheme brings several advantages to the domain of accurate, low-power asset tracking. First, the energy consumption of the tracked device can be kept relatively low since it only needs to be awake during a fraction of the ultrasonic chirp transmission. Second, chirps bring the advantage of pulse compression. They are signals with relatively long duration that not only increase the effective transmitted power for better SNR but also enable high compression in the time domain via cross-correlation. This results in both good target detection and ranging resolution. Additionally, chirp signals pose great resilience against multipath fading, which is omnipresent in typical indoor environments [33].

A well-considered choice of the chirp length $\Delta t_{\mathrm{chirp}}$, bandwidth Δf, and sample time window $\Delta t_{RX}$ at the tracked device is important to obtain a good ranging accuracy [30]:A large chirp bandwidth Δf positively impacts the ranging resolution. However, it is limited by the frequency response of the ultrasonic speaker and microphone. This work considers ultrasonic chirps in the range 18 kHz to 32 kHz, based on the frequency response of a commercial MEMS speaker.A small chirp length $\Delta t_{\mathrm{chirp}}$ would result in a high compression ratio. However, it also determines the maximum ranging distance $\Delta x_{\max}=v_{\mathrm{sound}}\;\cdot \;\Delta t_{\mathrm{chirp}}$.Increasing the sample time window $\Delta t_{\mathrm{RX}}$ at the tracked device improves the SNR and accuracy quadratically but, on the other hand, increases the energy consumption to perform a single ranging measurement.

## 3. Beamforming a Chirp Signal with Arrays

For the phased array, we opted for a uniform planar array (UPA), where the speakers are positioned in a 2D plane with equidistant interelement spacing. This provides a relatively easy design solution requiring low control complexity and allows beam steering in both the azimuth and elevation dimensions. Moreover, it can be practically mounted in a typical room environment. In a uniform phased array, the interelement spacing is generally set to a half-wavelength of the operating frequency. This means that the most optimal beam pattern will only be achieved at this design-operating frequency. When a chirp is transmitted over a certain frequency interval Δf, the beam pattern will change because of the following two physical considerations: (I) the wavelength of the transmitted signal changes relative to the fixed interelement spacing. Consequently, constructive and destructive interference of the acoustic waves will occur in other places. (II) If beam steering is implemented by means of phase shifts instead of true-time delays, the beam direction will change as a function of frequency. This concept is called beam squint. Consequently, to keep the main lobe directed toward the same direction while changing the signal frequency, the phase shifts between the loudspeakers need to be updated. The remainder of this section examines at what operating frequency the array should be designed to benefit the overall ranging accuracy.

Although uniform planar arrays (UPAs) are considered throughout the remainder of this paper, we first assume three uniform linear arrays (ULAs) as a baseline for this study. As a UPA can in fact be seen as multiple ULAs packed together, a linear configuration simply allows us to study the beam patterns in only one dimension. The three ULAs consist of 10 omnidirectional loudspeakers. Each of the three ULAs is designed with a different interelement spacing, which has been set toa half wavelength at the design frequencies of 15 kHz, 25 kHz and 35 kHz. Considering the propagation speed of sound in air (around 343 m/s), this corresponds to an interelement spacing of, respectively, 11.4 mm, 6.86 mm and 4.90 mm. A ULA at the design frequency of 25 kHz with $\frac{\lambda }{2}$ spacing has been shown to be practically feasible by means of 4.8 mm-by-6.8 mm MEMS transducers in [34]. To investigate the impact of chirp transmission on the beam pattern of each ULA, the transducers are excited with five different frequencies ranging between 15 kHz and 35 kHz. The target steering angle is set to an angle of 50°, chosen roughly in between the two steering boundaries, being the array broadside (0°) and the array axis (±90°). The calculations are performed in Matlab using the Phased Array System Toolbox.

We first focus on the ULA designed at the frequency of 25 kHz, shown in Figure 2b. When the signal frequency equals the array design frequency of 25 kHz, it is clear that the beam pattern is most optimal, with a plain main lobe and minimal side lobes. Lowering the signal frequency increases the beam width of the main lobe and reduces the number of side lobes. A larger main lobe width means that potentially more strong reflected waves exist that reach and interfere with the line-of-sight (LoS) signal at the tracked device, degrading the ranging accuracy. The same conclusion can be drawn for the higher signal frequencies. In this case, the width of the main lobe decreases with frequency, yet grating lobes appear in the beam pattern. Grating lobes, which are additional lobes with the same strength as the main lobe that result from the spatial aliasing effect between the signal wavelength and array interelement spacing, are undesired. More specifically, they appear when the element spacing is larger than $\frac{\lambda }{2}$ for the considered frequency. For this ULA configuration, designed at 25 kHz, grating lobes will emerge for frequencies above 25 kHz. From Figure 2b, it is clear that the direction of the grating lobe changes with signal frequency. This means that during chirp transmission, these lobes can sometimes align with certain specular reflection paths inside a room, causing multipath interference at the tracked device. However, their influence will only be temporary as their direction changes.

The beam patterns of the ULAs at the design frequencies of 15 kHz and 35 kHz are shown, respectively, in Figure 2a and Figure 2c. The results show that the ULA design comes with a trade-off regarding the interelement spacing. If the array is designed at the highest chirp frequency (Figure 2c), no grating lobes will appear in the beam pattern. However, the main lobe will have an overall larger beam width. This increases the chance that strong multipath components, originating from nearly the same direction as the LoS, will be picked up by the tracked device. Such a situation occurs, for example, when the tracked device is quite distant from the phased array and rather close to a wall. When the array interelement spacing is optimized for the lowest chirp frequency (Figure 2a), one or multiple grating lobes will be present for a longer time during chirp transmission, but the main lobe beam width is overall smaller. In this case, the tracked device could pick up strong multipath components from different directions. However, as the grating lobes are shifting during chirp transmission, only parts of the chirp signal will interfere with the LoS sporadically. This could mean that the correlation process in the ranging scheme (Figure 1) will be overall less negatively affected. In conclusion, a high array design frequency, which brings along a larger beam width, could pose a larger threat to the ranging performance. However, no definitive advantage can be attributed to one specific case. To strike a balance between main lobe beam width and grating lobes, we aligned the array design frequency with the chirp center frequency.

## 4. Optimizing the UPA Design: A Room Evaluation

The interelement spacing determines only a part of the array design. After all, the envisioned UPA still exhibits other degrees of freedom that can impact the beam pattern of the array, namely the following.

Number of speakers. Adding more speakers to the array will increase the total array gain and reduce both the main lobe width and side lobe levels. This positively affects the SNR and thus the ranging accuracy, yet it comes at the expense of increased system complexity and cost.Array configuration. Changing the horizontal and vertical distribution of the speakers in the UPA makes it possible to adapt the beam pattern in both dimensions. A reduced main lobe width in one dimension can, for example, be traded for an increased width in the other dimension if it favors the situation.

This section investigates the effect of both array design variables on the ranging performance by means of simulation. Two considerations are taken into account. First, the ranging accuracy will be influenced by the environment in which the system is deployed. As this work targets the positioning of devices in typical indoor environments (e.g., stores), shoe-box-type rooms will serve as a baseline throughout these simulations. Second, the investigation focuses on (quasi-)static devices. Mobile devices would require a continuous redirection of the main lobe. In the case of narrow main lobes, fast and accurate beam pattern adjustments are needed. This could potentially affect certain design trade-offs that are not applicable in the case of static devices. The simulations are performed in Pyroomacoustics [35], a Python package for audio signal processing in indoor applications.

Specifically, the following two research questions are investigated: (I) Which UPA topology, different in configuration and amount of speakers, leads to the best ranging accuracy considered over an entire shoe-box room? (II) How does the UPA beamforming approach perform in terms of ranging accuracy compared to a traditional, single-speaker setup?

### 4.1. Simulation Setup

This section starts off with the first research question: Which UPA topology offers the best ranging accuracy in typical room environments? To avoid optimization for only one specific room, three shoe-box rooms with different dimensions are considered in the simulations. The rooms are presented in Figure 3, and the specific dimensions are shown in Table 1. Room 1 has a nearly square ground plane, while room 2 is a typical bar-shaped room with slightly different lengths and widths. Room 3 represents a hallway, where the length is significantly larger than its width. All rooms have the same height of 3 m. Like the room’s dimension, the location of the UPA in the room can also affect the overall ranging performance. To make an abstraction in a similar way, twelve test locations (three rows, four columns) have been uniformly selected in one quadrant of the front wall, as shown in Figure 3. The locations are actually moved 5 cm from this wall to mimic real mounting. Within this quadrant, both the left column and top row of speakers are at 20 cm distance from the left wall and ceiling, respectively. The rightmost column and bottom row speakers are located on the center lines of the rectangular front wall, respectively, at $\frac{L}{2}$ and 1.5 m. The other test locations are chosen uniformly within this box of speakers, with the coordinates shown in Figure 3.

Thirteen UPA configurations were evaluated, listed in Table 2. To ensure that the considered UPA configurations are actually practically feasible, we used the ultrasound phased array from [34] as a baseline design. In the previous section, we showed that designing loudspeaker arrays at high frequencies results in very small interelement spacings (<1 cm). Consequently, we are very limited by the dimensions of the speaker transducers. The baseline design uses MEMS transducers that measure 4.8 mm-by-6.8 mm and have a working frequency response in the (near-)ultrasonic range of 18 kHz to 32 kHz. Taking both the conclusions from the previous section and the working dimensions of the reference array into account, the interelement spacing of all UPA configurations are tuned to $\frac{\lambda }{2}$ at the center frequency of 25 kHz. The directivity pattern of a single speaker is set to a backbaffled subcardioid, in which acoustic transmissions at the rear are attenuated by means of an absorber. In this way, strong reflections on the wall behind the array can be prevented. In addition, we opted to limit the number of speakers to 20 at maximum, to limit the cost and complexity. The UPA configurations listed in Table 2 are presented in the format A × B. A represents the number of speaker rows, placed in parallel with the length of the room. B represents the amount of speaker columns, placed in parallel with the height of the room. A 2 × 10 configuration, for example, has 2 rows of 10 speakers on top of each in other in the height direction.

To assess the overall ranging accuracy of a specific UPA configuration, we follow the simulation flow as clarified in Algorithm 1. Each shoe-box room is filled with virtual microphones (representing microphones on tracked devices), spaced 20 cm from each other in all dimensions. This results in 15,246, 13,650 and 9240 microphones in rooms 1, 2 and 3, respectively. A single simulation involves placing the UPA in one of the twelve selected wall positions. Ranging measurements are then performed between all microphones and the UPA using the acoustic signaling approach from Section 2. As an acoustic signal, we chose a chirp with linearly decreasing frequency from 32 kHz to 18 kHz over a time interval of 30 ms. Both this chirp duration and the room dimensions have been chosen such that the chirp signal can at least propagate to every point in the room. The energy-absorption coefficients of the walls, floor and ceiling are distinctively set to a constant factor of 0.3, representing a room covered in fiberboard in the sound frequency domain [36]. The maximum reflection order is set to 5. From simulations, we noticed that higher-order reflections did not affect the ranging performance. Next to wall absorption, an acoustic signal is also attenuated through air absorption, modeled as $e^{-\alpha d}$, where α is the air-absorption coefficient and *d* is the distance that the acoustic signal has traveled. At a temperature of 20°, relative humidity of 30% and frequency of 25 kHz, the air-absorption coefficient is set to 0.0875 Np/m [37]. The acoustic reception window of the tracked device is fixed at 1 ms to keep its energy consumption low. The sampling frequency of the simulation environment is set to 1 MHz, but the microphone signal is resampled at 400 kHz to mimic a low-power, resource-constrained microcontroller device. A standard cross-correlation is performed between the received microphone signal and the originally transmitted chirp to calculate the distance. No additional noise is added to the microphone signal before the cross-correlation calculation. An in-depth study on how additive white Gaussian noise affects the accuracy and precision of the ranging measurements is presented in [30]. When the ranging measurement is conducted, the main lobe of the UPA is always steered perfectly toward the microphone location. Hence, the simulation shows the best possible scenario. In practice, the relative azimuth and elevation angle must be obtained through a different strategy, and an error may affect optimal beamforming. One could, for example, fall back to an angle of arrival (AoA) setup using the RF signals obtained from the tracked device. Once all distance measurements have been conducted, the errors between the real UPA–microphone distances and the simulated distances can be calculated. The same simulation is repeated for all UPA configurations in every targeted wall position.
**Algorithm 1** Simulation flowSet up simulation environment based on settings (see Table 3).**for** room *r* in rooms (see Figure 3) **do**    Fill *r* with virtual microphones, spaced 20 cm in x, y and z.    **for** UPA in configurations (see Table 2) **do**        **for** UPA position p=0 to 11 (see Figure 3) **do**            Set UPA at position *p* in room *r*.            **for each** virtual microphone *i* in room *r* **do**                Set UPA delays for beamforming towards microphone *i*.                Perform RF–acoustic ranging measurement between UPA and microphone *i* (see Algorithm 2).                Calculate the distance error with respect to the theoretical distance.                Save the distance error for the UPA–microphone pair.            **end for**        **end for**    **end for****end for**Calculate the P50 and P95 distance errors, for each UPA at position *p* in room *r* separately.

**Algorithm 2** RF–acoustic ranging measurement
Calculate the room impulse response (RIR) for the full UPA and microphone *i*.Obtain the captured signal at microphone *i* by convolving the RIR and chirp.Extract part of the captured microphone signal in the time interval [30 ms to 31 ms].(*This mimics the tracked device’s reception window of 1 ms after RF wake-up. Refer to $\Delta t_{RX}$ in Section 2.*)Correlate the extracted microphone signal and chirp to calculate the TOF.Calculate the distance = TOF ·$v_{sound}$


### 4.2. Results

In this section, we evaluate the ranging performance of the UPA configurations in the different setups. In this regard, both the P50 and P95 ranging errors are calculated and presented. This section mainly gives an overview of the initial observations, while the section hereafter provides a more in-depth discussion of the findings.

The simulation results concerning the P50 and P95 ranging errors are shown, respectively, in Figure 4 and Figure 5. Both figures show 12 subplots, each representing one of the tested UPA positions. The subplot in the top left of Figure 4 and Figure 5 (position 2) corresponds to the position at the top left corner of the wall in Figure 3. Each subplot displays the P50/95 ranging errors for the thirteen tested UPA configurations. To present the different configurations on a single axis (x-axis) in an intuitive way, each one is represented by an array configuration factor. This factor represents the number of speakers in the largest UPA dimension. A negative array configuration factor indicates that most speakers in the UPA are mounted above each other in the height direction of the room, while a positive value indicates that most speakers are positioned horizontally in the length. The array configuration factor corresponding to each array is presented in more detail in Table 2. As the 6 × 3 and 6 × 2 array designs have the same array configuration factor of −6, they can be further differentiated based on the number of speakers. A larger circle in Figure 4 and Figure 5 corresponds to more speaker elements, making it possible to distinguish both array configurations. The 4 × 4 UPA configuration forms an exception and is represented by an array configuration factor of 0. The room in which the ranging performance is evaluated is indicated by the same room color as in Figure 3.

We first focus on the P50 ranging errors depicted in Figure 4. When the UPAs are mounted toward the middle of the wall rather than near the sides (positions 3, 4, 6, 7, 9, 10), a similar trend can be observed in the ranging performance. The vertical array configurations show the lowest P50 ranging errors, regardless of the room. When a more horizontal array configuration is used, the ranging error becomes larger. This is even more so for room 1, with the nearly square ground plane, compared to room 3, which is a hallway-like environment. A hypothesis for this observed trend can be found in the dimensions of the rooms. Since the heights of the rooms are relatively small compared to their lengths, and thus, shortly following, fairly strong reflections could interfere with the LoS, limiting the reflections in this dimension could be most important. After all, the vertical arrays have a small beam width in the height direction. If we compare the 6 × 2 and 6 × 3 vertical array configurations, we can state that the extra speaker column in the 6 × 3 array brings hardly any benefit and that the number of speakers in the vertical dimension dominates. This again supports the previous hypothesis that avoiding reflections in the height dimension is more important than in the horizontal dimension. Overall, a similar conclusion can be drawn for UPA positions 5, 8, and 11 near the ceiling. Again, the vertical array configurations perform best, showing an equally low-ranging error as the middle positions. While the horizontal arrays perform only slightly worse, the room dependence has now disappeared. The trend reverses when the UPAs are mounted in positions 0 to 2, close to the left wall. In room 1, the P50 ranging error remains more or less the same for almost all UPA configurations. In rooms 2 and 3, on the other hand, the horizontal arrays give rise to the lowest error. The vertical arrays perform gradually worse when the room stretches out in the horizontal dimension. The fact that the vertical arrays perform worse in these UPA positions could be explained by their beam pattern. As these arrays have a large beam width in the horizontal dimension, fairly strong reflections can occur on the west (left) wall of the room, possibly complicating the ranging measurement. Furthermore, when the room stretches out in the length dimension, many virtual microphones are located progressively farther from these UPA positions. As many virtual microphones are still located close to positions 5, 8 and 11, this may explain why no troubling ceiling reflections can be observed for the horizontal array configurations. It is to be noted that, in this case, we are only considering the 50% best-performing microphone locations. Overall, placing the UPAs in positions 0 to 2 leads to a higher P50 ranging error compared to all other positions. When we compare the 2 × 6 and 3 × 6 horizontal array configurations, the number of horizontal speakers again prevails over the extra row of speakers in the 3 × 6 configuration.

The P95 ranging errors, presented in Figure 5, show how the array configurations perform over the entire rooms. Zooming in on the middle UPA positions (3, 4, 6, 7, 9, 10) we can observe that a low P95 ranging error can be achieved by placing the speakers as much as possible in either the horizontal or vertical dimension, rather than in between. This could be explained by the fact that a narrow beam is created in at least one dimension, preventing certain multipath components from propagating to the tracked device. The vertical array configurations still have a slightly lower P95 ranging errors in most of these UPA positions regardless of the room, except in room 3 at positions 3 and 4. In this case, the horizontal array configurations tend to outperform the other configurations. At positions 5, 8 and 11 near the ceiling, yet distant from the left wall, there is a clear preference shift toward the vertical UPA configurations in all rooms. An extra speaker column in the 6 × 3 configuration compared to the 6 × 2 does not necessarily provide a significant gain. As in the case of the P50 ranging errors, a reversed trend can be observed for positions 0 to 2. In this case, the horizontal array configurations lead to the lowest P95 ranging errors due to the nearby west wall. The vertical array configurations again perform significantly worse in hallway-like room 3 compared to room 1.

In conclusion, the lowest P50 and P95 ranging errors are achieved when the UPA is not mounted near any walls. This does not necessarily have to be the center of the mounting wall. In these cases, the 10 × 2 vertical array configuration performs best overall, with a P50 ranging error around 3 cm and P95 ranging error below 30 cm. This stands in significant contrast with UPA positions 0 to 2, where significantly larger P50 ranging errors can be observed for the vertical arrays, potentially resulting from the nearby, reflecting west wall. The fact that high ranging accuracies can be achieved with just one array at one location shows the enormous potential of the proposed system for many applications. Moreover, it hints that adding more phased loudspeaker arrays may yield strong results for positioning a device in 3D, making it highly competitive with current other technologies [38]. Depending on the application and targeted ranging accuracy, one could potentially opt for a vertical UPA with fewer speaker elements. The 8 × 2 and 6 × 2 UPA configuration often leads to only slightly worse ranging errors compared to the 10 × 2 UPA, but the design complexity and cost can be reduced.

## 5. Results Discussion

In this section, we study the observations and hypotheses made above in more detail. In Section 5.1, we focus on the observations made for UPA positions 0 to 2. In Section 5.2, we investigate how the vertical configurations manage to outperform the horizontal ones in the best-performing positions, 3 to 11.

### 5.1. UPA Positions 0–2

In the previous section, we observed that vertical arrays performed worse when they were mounted in UPA positions 0 to 2, closest to the left wall, compared to all other positions. The horizontal arrays, on the other hand, were only slightly affected. A plausible hypothesis was pitched that attributed these results to the larger beam width of vertical arrays in the horizontal direction. In this way, strong reflections on the nearby left wall can easily interfere with the LoS.

To further investigate these observations, we plotted the beam pattern of the 2 × 10 horizontal and 10 × 2 vertical array spatially at the frequency of 25 kHz, as shown in Figure 6. The UPAs are mounted in position 4 of room 2 to create some field resolution towards the west wall (left wall in the figures). The beams are directed toward the middle of the east wall (right wall in the figures) since many virtual microphone locations lie on the right side of the UPAs. As expected, the 2 × 10 UPA has a high resolution in the azimuth direction, i.e., in the plane z = height of the UPA, but only small resolution in the elevation direction, i.e., the angular height with respect to the aforementioned z-plane. The 10 × 2 array, on the other hand, shows a good beam steering resolution in the elevation direction, but little in the azimuth direction. Notably a strong side lobe is directed toward the west wall in case of the vertical 10 × 2 UPA. A side lobe is also present at the 2 × 10 horizontal UPA, but not as prominent. When the 10 × 2 UPA is positioned closer to the west wall, the side lobe’s ultrasonic waves would quickly follow those of the main lobe, potentially causing trouble to the ranging performance. To test this hypothesis, the west wall was made completely acoustically absorbent. The ranging performance was again evaluated for all array configurations, but only at position 1 in room 2. The P50 and P95 ranging errors are shown in a light blue color in Figure 4 and Figure 5, respectively. As can be seen in both figures, the P50 and P95 ranging errors drop drastically for the vertical array configurations, while those of the horizontal ones are fairly little affected. The vertical UPAs now takes again the upper hand in providing the best ranging performance. This clearly demonstrates that the placement of the 10 × 2 array near a wall must be avoided.

### 5.2. UPA Positions 3–11

While the preceding simulations indicated which UPA configurations led to a low ranging error, they did not show how the vertical array configurations managed to outperform the horizontal arrays for the other positions (3 to 11). To gain more insights in this regard, the spatial distribution of the ranging errors over the entire rooms is examined. Two examples are shown in Figure 7a,b, where a 2 × 10 and 10 × 2 array are placed at UPA position 7 of room 2, respectively. The UPAs are not visible in the figures as they are located on the left rear wall of the rooms shown (room 2 from Figure 3 has rotated a little over 90° clockwise around its ground plane’s normal). The simulated ranging error in every microphone location is indicated using a color. When both error distributions are compared, larger ranging errors can be observed near the walls for the 2 × 10 horizontal array case. Moreover, the same observations can be made when the arrays are placed in UPA positions 3 to 11 (not shown). Since the 10 × 2 array has a narrow main lobe in the vertical direction, this indicates that reflections through the floor and ceiling potentially play a significant role in the ranging performance. After all, the height of the room is small compared to the length and width of the room, so the little-attenuated first-order reflections through floor and ceiling and their derivatives could easily interfere with the LoS.

To further investigate the impact of floor and ceiling reflections on the ranging errors, room 2 and UPA position 7 are again considered as the test setup. As before, the array configurations from Table 2 are placed in the targeted UPA position, and the ranging errors to all microphone points are simulated. However, this time, the energy-absorption coefficient of both the floor and ceiling are adjusted between 0 and 1. More specifically, five energy-absorption coefficients are evaluated, 1.0, 0.7, 0.4, 0.1, and 0.01, ranging from total absorption to almost full reflection. In this way, the reflections on these surfaces can be controlled. The energy-absorption coefficients of the walls are kept constant at 0.3.

Figure 8 shows the P95 ranging errors for the different arrays’ configurations and energy-absorption coefficients of floor and ceiling. From these results, it can be seen that the 10 × 2 and 8 × 2 vertical UPAs are able to achieve a low (<0.4 m) P95 ranging error, even with a highly reflective floor and ceiling surface. The horizontal and also vertical arrays with fewer speaker elements, on the other hand, show a significant increase in the P95 ranging error at low energy-absorption coefficients. When the floor and ceiling are (almost) completely absorbing, the horizontal array configurations barely outperform the vertical configurations. This indicates that the reflections through the vertical walls are overall less impactful. This was to be expected, as the initial reflections on the vertical walls should be mainly excluded in the case of the horizontal configurations due to their narrow main lobe in the azimuth direction. Moreover, as the walls are further apart compared to the floor and ceiling, multipath signals propagating through these paths will experience higher attenuation due to the larger distance and longer air absorption experienced.

Similarly, it can be tested whether the preference shifts towards horizontal configurations when the side walls, i.e., the walls left and right of the mounting wall, are put closer together relative to the height of the room. A room with length 3.16 m, width 8.0 m and height 5.09 m is used as a test setup (not shown). The UPA designs are mounted 5 cm in front of the shortest wall, at a distance of 1.52 m from one sidewall and 2.4 m in height, which is close to its center. Again, all previous UPA configurations are tested, and the energy-absorption coefficients of these side walls are changed between 0 and 1. The energy absorption values of the floor, ceiling and all other walls are kept constant at 0.3.

Figure 9 shows the P95 ranging error for the different array configurations and energy-absorption coefficients of the side walls. It can be seen that, in fact, lower ranging errors are achieved with horizontal array configurations, at least for the energy-absorption coefficients of the side walls, between 0.01 and 0.7. When the side walls are completely absorbent, the vertical array configurations again take the upper hand.

In conclusion, simulations have shown that the smallest room dimension determines the UPA configuration to a great extent. For typical room dimensions, where the length and width are considerably larger than the height, the 10 × 2 vertical UPA gave the lowest ranging errors overall. This shows that the largest UPA dimension best coincides with the smallest room dimension to obtain the lowest ranging errors over the entire room, at least for similar room shapes. Furthermore, we have established that even when the floor and ceiling became highly reflective, the 10 × 2 UPA still managed to keep the P95 ranging error in the same order of magnitude.

### 5.3. Phased Arrays versus Single Speakers

This section focuses on the second research question: does a phased speaker array perform better than a single-speaker system for indoor ranging, and if so, to what extent? After all, a phased array raises the design costs and complexity for the anchors and must therefore bring a sufficient performance gain in order to be practically viable. However, multiple individual loudspeakers may be needed to achieve the same performance as one phased loudspeaker anchor. As a test environment, we choose room 2 from Figure 3 with a moderate energy-absorption coefficient of 0.3 for all surfaces. In this simulation, the 10 × 2 vertical UPA is compared to a single speaker, since it came out best for this environment according to the previous assessment. The beam pattern of the single speaker is equal to that of the speakers in the UPA. Both the 10 × 2 array and single speaker are mounted in the targeted UPA positions as shown in Figure 3 to create some spatial diversity. The same ranging signals and settings are used as in previous simulations.

Figure 10 shows the cumulative distribution function (CDF) plots of the ranging errors for both the 10 × 2 vertical UPA and the single speaker case in all twelve tested transmit positions. The plots are based on the distance errors simulated in all microphone locations spread across the entire room. In addition, the P50, P80, and P95 ranging errors are listed to allow a simple numerical comparison. The results show that in all tested transmit positions, the UPA significantly outperforms the single speaker setup. The 10 × 2 array is able to lower the P50 ranging error with a factor of 3 to 5 at all positions. In position 9, for example, the P50 is brought down from 11 cm to only 3 cm. The ranging performance of the UPA especially excels when the P80 and P95 are compared. When looking at position 9, which shows the lowest ranging errors, both the P80 and P95 are reduced with a factor of 8 to 9. For some other UPA positions, the P80 and P95 errors even drop a spectacular entire order of magnitude. This shows that the UPA is able to reduce many outliers very effectively through its narrow ultrasound beams.

### 5.4. Discussion Conclusions

The obtained accuracies underscore the huge potential of the presented phased array ranging system, especially, considering the fact that these results were obtained using only a single loudspeaker array in one location. This indicates that incorporating additional phased loudspeaker arrays has great potential to enable highly accurate and precise 3D positioning across entire room spaces. Furthermore, the utilization of cost-effective commercial off-the-shelf (COTS) components for this system, along with the typical low energy consumption of MEMS microphones at the tracked device side, makes it a compelling choice for positioning energy-neutral devices [39,40]. Overall, these factors position it as a highly competitive alternative to existing state-of-the-art solutions.

In order to direct the acoustic signals towards a tracked device, both azimuth and elevation angles are needed. Abstraction has been made in this work on how this angle information is obtained, especially during initial access. Nonetheless, an RF direction of arrival could be employed to extract this information from the device’s communication signals [32]. How the accuracy and precision of such a system affect the ranging performance of the proposed RF–acoustic system should be further investigated in future work.

Understanding which limitations can be expected in the performance gain of the proposed method poses a rather complex question. The following three factors will at least have a major influence in this regard.

The loudspeaker array design. Increasing the number of speaker elements would further decrease the beam width of the main lobe. Consequently, fewer multipath components would be able to interfere with the LoS signal, further improving the acoustic ranging measurement in more room locations. On the other hand, mobile devices bring another level of complexity to the system since a smaller main lobe beam width will require faster redirection.The beamforming feedback system. As mentioned before, we idealized the acoustic beamforming such that the ranging signals were perfectly directed toward the tracked device. In practice, the measured azimuth and elevation angles of the tracked device with respect to the phased loudspeaker beacon will be subjected to errors, affecting the ranging measurements.The environment. We only considered shoe-box rooms in these simulations. Adding objects or introducing non-shoe-box environments would significantly increase the simulation complexity. Consequently, practical measurements in real environments should provide further insights in this regard.

## 6. Conclusions and Future Work

In this work, we presented a novel hybrid RF–acoustic ranging system based on phased loudspeaker arrays for indoor localization applications. The design of a uniform planar array (UPA) up to 20 speakers was optimized to achieve the best ranging performance in medium-sized shoe-box rooms. The RF–acoustic ranging principle used in this work was based on the state-of-the-art ranging scheme from [30], relying on the pulse compression of ultrasonic chirps. It enables both robust and energy-efficient TOF measurements, making the system viable for ultra-low-power or even energy-neutral tracked devices. Extensive ranging simulations, testing eleven UPA configurations over more than 9000 tag locations, showed that a 10 × 2 vertical speaker array led to the overall lowest ranging errors in typical room environments, i.e., when the length and width of the room are considerably larger than its height. In these cases, a P50 ranging error of around 3 cm and P95 ranging error below 30 cm were achieved. Additional simulations showed that the largest UPA dimension should coincide with the smallest room dimension to achieve the best ranging performance. When a 10 × 2 vertical UPA was compared to a conventional single speaker ranging system, the P80 and P95 ranging errors could be lowered with factors between 8 and 9. This showed that the UPA is able to eliminate many outliers that are present in the case of the single-speaker ranging system.

We see great opportunities for a wide range of new follow-up studies. A practical measurement campaign should be conducted to validate the presented simulation results in a real-life setup. Hereafter, the presented ranging system can be expanded to a full-fledged 3D positioning system. Moreover, only static tag devices were considered. Consequently, additional design guidelines can become important when moving targets are considered.

## Figures and Tables

**Figure 1 sensors-23-07997-f001:**
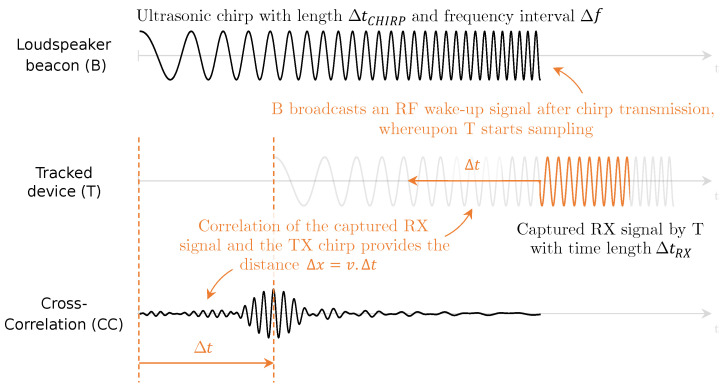
Visual representation of the low-power RF–acoustic ranging approach from [30]. Loudspeaker beacon *B* initiates a ranging measurement by broadcasting a linear ultrasonic chirp with length $\Delta t_{chirp}$ over a frequency interval Δf. At the end of the chirp transmission, beacon *B* sends out an RF signal, signaling tracked device *T* to immediately wake up and sample a part of the still propagating chirp signal over a period $\Delta t_{RX}$. Depending on its distance to loudspeaker beacon *B*, the tracked device’s fragment will contain a specific frequency interval of the original chirp. Cross-correlation is used to find out which frequency interval was seen by tracked device *T*, resulting in a propagation delay and consequently the distance.

**Figure 2 sensors-23-07997-f002:**
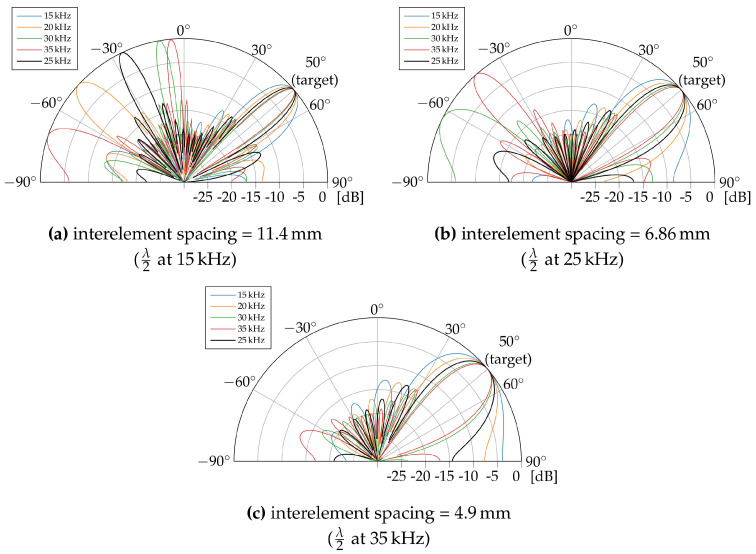
Beam patterns of the three ULA configurations with different interelement spacing. The target steering angle is set to 50${}^{\circ }$. The color of each beam pattern indicates the signal frequency.

**Figure 3 sensors-23-07997-f003:**
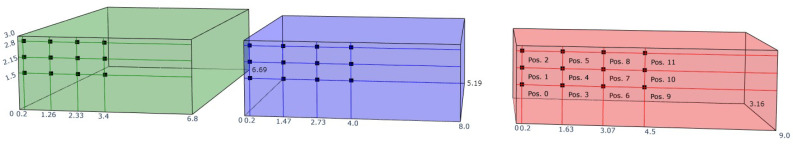
Shoe-box rooms considered in the simulations, referred to as room 1 (green), room 2 (blue), and room 3 (red). The rooms have the same height, but different floor sizes. Their specific dimensions can be found in Table 1. The black markers indicate the positions where the uniform planar arrays (UPAs) are mounted. The heights of the three marker rows are the same for all rooms, but they are only indicated for room 1.

**Figure 4 sensors-23-07997-f004:**
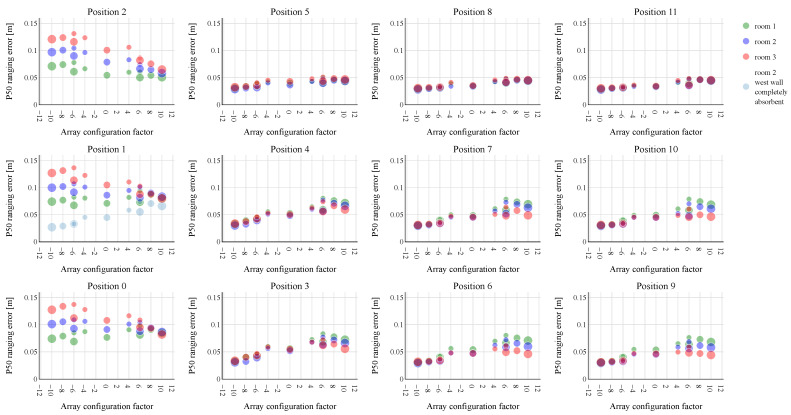
P50 ranging errors obtained for the different UPA configurations from Table 2. The subplots represent the twelve UPA positions in the rooms, as shown in Figure 3. The colors of the markers indicate the shoe-box rooms considered. In essence, the lowest P50 ranging errors are obtained by using a 10 × 2 vertical UPA configuration, not mounted near walls. When the UPAs are mounted near the west wall (positions 0–2), the horizontal array configurations present the lowest P50 ranging errors. By making the west wall completely absorbent, we confirmed that the vertical array configurations previously suffered from a strong, interfering reflection on this wall. This strong reflection is spatially shown in Figure 6.

**Figure 5 sensors-23-07997-f005:**
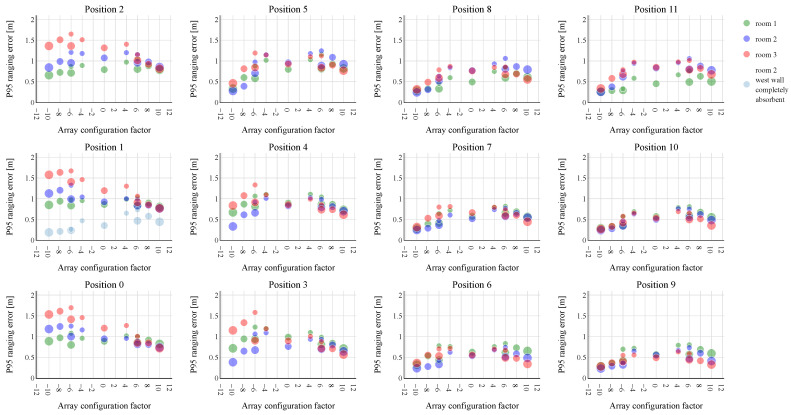
P95 ranging errors obtained for the different UPA configurations from Table 2. The subplots represent the twelve UPA positions in the rooms, as shown in Figure 3. The colors of the markers indicate the considered shoe-box rooms. In essence, the lowest P95 ranging errors are obtained by using a 10 × 2 vertical UPA configuration, not mounted near walls. When the UPAs are mounted near the west wall (positions 0–2), the horizontal array configurations show the best ranging performance over the entire room. By making the west wall completely absorbent, we confirmed that the vertical array configurations previously suffered from a strong, interfering reflection on this wall. This strong reflection is spatially shown in Figure 6.

**Figure 6 sensors-23-07997-f006:**
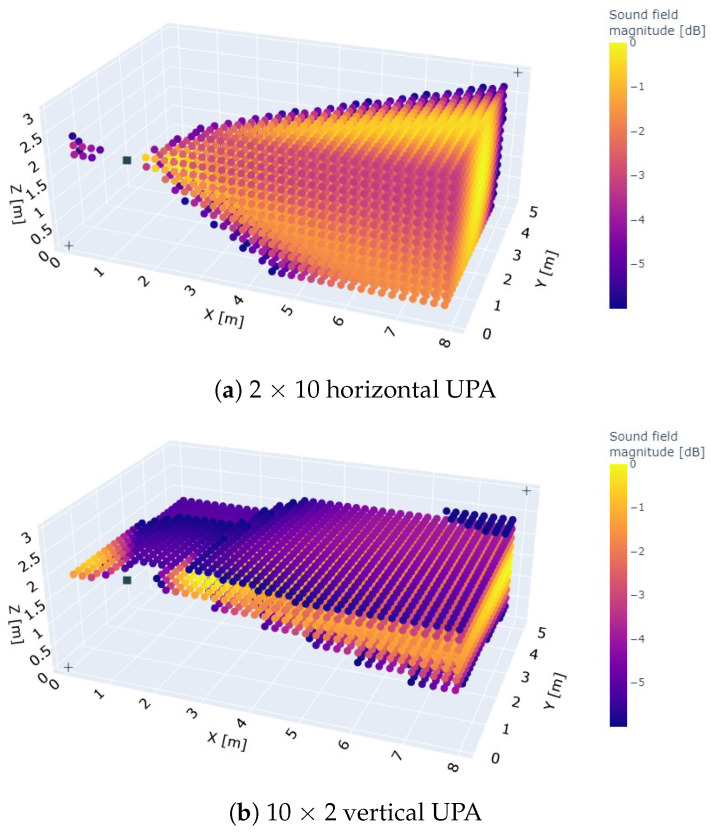
Spatial beam pattern plot of the (**a**) 2 × 10 array and (**b**) 10 × 2 array when beamforming a 25 kHz sine wave toward the east wall in room 2. A strong side lobe is directed towards the west wall in the case of the 10 × 2 vertical UPA. This side lobe is also present for the 2 × 10 horizontal UPA, but not as prominent.

**Figure 7 sensors-23-07997-f007:**
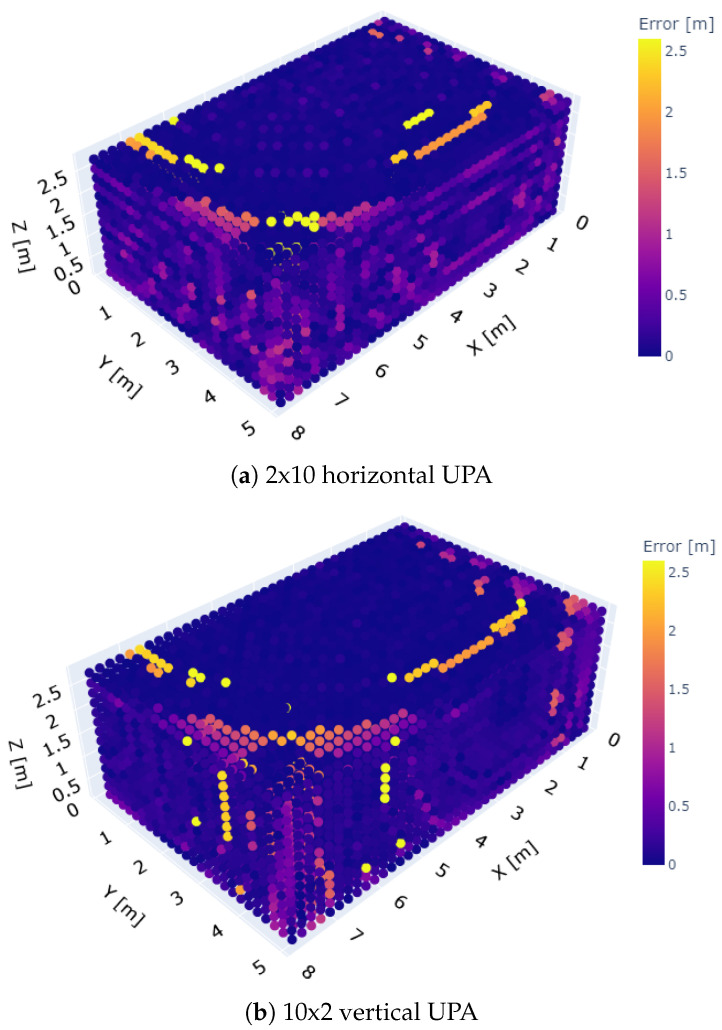
Room distributions of the ranging errors obtained at UPA position 7 for the (**a**) 2 × 10 array and (**b**) 10 × 2 array configuration. Overall, larger ranging errors can be observed near the walls for the 2 × 10 horizontal UPA compared to the 10 × 2 vertical UPA. This indicates that reflections through the floor and ceiling potentially play a significant role in the ranging performance.

**Figure 8 sensors-23-07997-f008:**
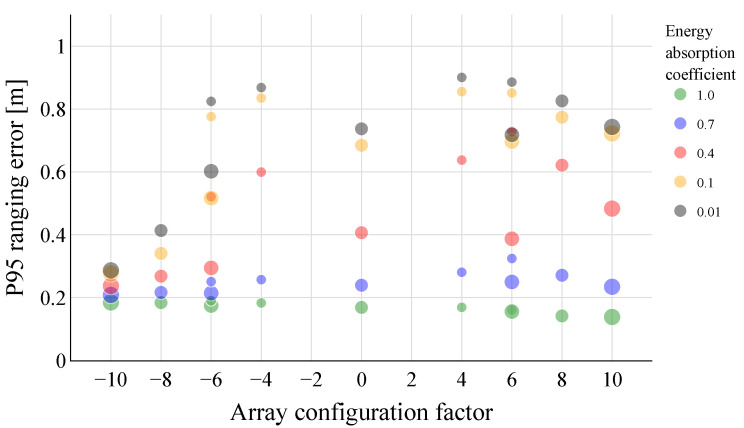
P95 ranging errors were obtained for the different UPA configurations at UPA position 7 in room 2. The energy-absorption coefficients of the floor and ceiling are changed between 0 and 1, while those of the walls are kept constant at 0.3. For high energy-absorption coefficients, the horizontal UPAs barely outperform the vertical ones. This indicates that the reflections through the vertical walls are overall less impactful. Even in the case of a highly reverberant floor and ceiling, the 10 × 2 and 8 × 2 vertical UPAs manage to keep the P95 ranging error below 0.4 m.

**Figure 9 sensors-23-07997-f009:**
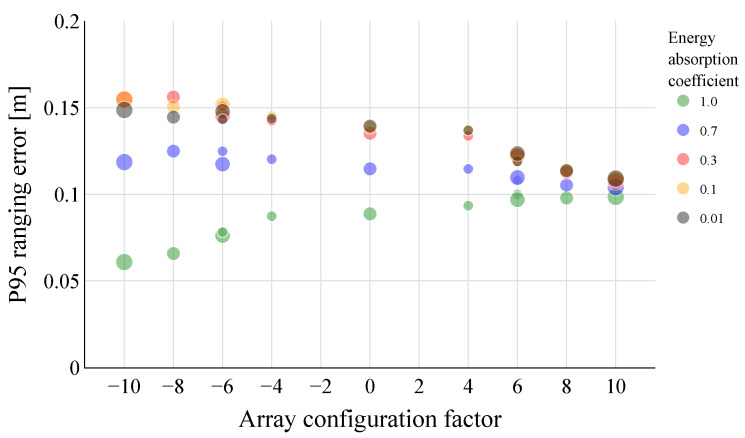
P95 ranging errors obtained for the different UPA configurations in the room with length 3.16 m, width 8.0 m and height 5.09 m. The energy-absorption coefficient of the east and west sidewalls are changed between 0 and 1, while those of the other sidewalls, floor and ceiling are kept constant at 0.3. Overall, the horizontal UPAs manage to outperform the vertical array configurations. Considering the small width of the room, a small beam width in this dimension proves to be key to achieving low P95 ranging errors.

**Figure 10 sensors-23-07997-f010:**
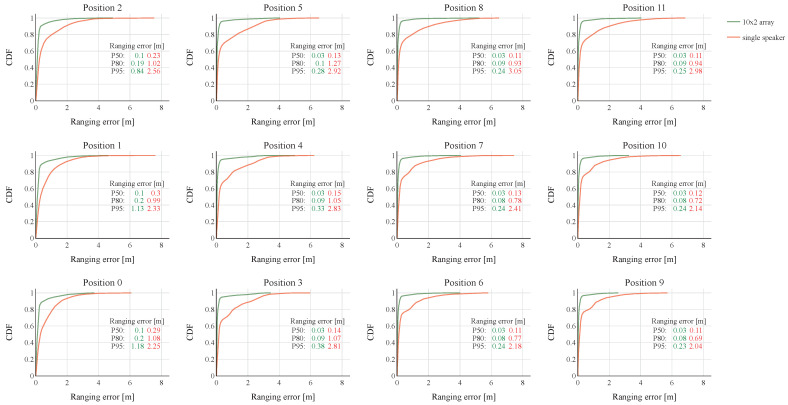
Cumulative distributions functions of the room ranging errors for both the 10 × 2 vertical UPA and single-speaker case. The twelve subplots represent the specific transmit locations.

**Table 1 sensors-23-07997-t001:** Specific dimensions of the shoe-box rooms in Figure 3.

	Room 1	Room 2	Room 3
Length (L) (m)	6.8	8.0	9.0
Width (W) (m)	6.69	5.19	3.16
Height (H) (m)	3.0	3.0	3.0

**Table 2 sensors-23-07997-t002:** Tested UPA configurations, presented in an A × B format. A represents the number of speaker rows, placed in parallel with the length of the room. B represents the number of speaker columns, placed in parallel with the height of the room. A 2 × 10 configuration, for example, has 2 rows of 10 speakers on top of each in other in the height direction. Each array configuration is also represented by an array configuration factor, which is defined by the number of speakers in the largest UPA dimension. A negative array configuration factor indicates that most speakers in the UPA are mounted above each other in the height direction of the room, while a positive value indicates that most speakers are positioned horizontally in the length.

Configuation	10 × 2	8 × 2	6 × 2	6 × 3	4 × 3	4 × 4	3 × 4	3 × 6	2 × 6	2 × 8	2 × 10
Array configuration factor	−10	−8	−6	−6	−4	0	4	6	6	8	10

**Table 3 sensors-23-07997-t003:** Simulation and room settings.

Variable	Signal/Value
Source signal	Linear chirp from 32 kHz to 18 kHz over 30 ms
Wall energy absorption	0.3
Room reflection order	5
Air-absorption coefficient	0.0875 Np/m
Microphone reception window	1 ms
Simulation sample rate	1 MHz
Microphone sample rate	400 kHz

## Data Availability

Data sharing not applicable.

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
