# Peer review of "Accurate and Low-Power Ultrasound–Radiofrequency (RF) Indoor Ranging Using MEMS Loudspeaker Arrays"

_sensors, 2023, doi:10.3390/s23187997_

Round 1
Reviewer 1 Report
The study focuses on an accurate and low power ultrasound-RF indoor ranging using MEMS loudspeaker arrays. In particular, the authors optimize the design of a low-cost UPA through simulation to achieve the best ranging performance using ultrasonic chirps, and compare the ranging performance of this optimized UPA configuration to a traditional, single loudspeaker system. The study is relevant and poses good significance. However, revisions are required to improve the quality of the paper before publication.
1. The related work section is almost missing in this work. Only a few related works were referred to in section 1. While the work has addressed an important problem, the underlying problem is not adequately motivated. I recommend that the related work section should be added. The authors should explore more related work to better motivate the problem being addressed. In particular, the authors should identify the gaps in related works and demonstrate how the current study has filled the gaps.
2. I would prefer to see an experimental setting to complement the validation of the projected simulation results and practical comparison presented.
3. An algorithmic presentation of the simulation process is required to ease research reproduction.
4. The limitation of the proposed method requires elaboration.
5. I would suggest that the conclusion should be more concise. The parts involving citations may be taken to the discussion section.
Minor English editing is required.
Author Response
Dear reviewer,
Please find our responses to your comments in the attachment.

Reviewer 2 Report
With interest we were reading the paper and we have the following critical comment:
1. The research is based on the preceding work of van der Perre.
Readers not familiar with that work have problems in reading the current paper. The work of van der Perre should be more summarized. Presenting Fig 1 is insufficient.
2. There is some recent work in Remote Sensing 2023 which should be discussed
Hybrid Indoor Positioning System Based on Acoustic Ranging and Wi-Fi Fingerprinting under NLOS Environments Zhengyan Zhang 1 , Yue Yu 2,*, Liang Chen 3 and Ruizhi Chen. This paper has also a huge number of related works, some of them should be discussed in the current paper.
3. The paper has a section Contributions. But the research questions as stated in 4.1 and 4.4 should be stated at the beginning.
4. Most of the experimental results are stated without any explanation for example conclusions in line 375 etc.
5. It was stated that the work of van der Perre was used as a benchmark, that needs more explanation for ex. in the conclusion line 476
Author Response
Dear reviewer,
Please find our responses to your comments in the attachments.

Round 2
Reviewer 1 Report
The authors have addressed my earlier comments, remaining only the experimental setting which they claimed is not possible at this time.
Minor English editing.